# A Comprehensive Review of miRNAs and Their Epigenetic Effects in Glioblastoma

**DOI:** 10.3390/cells12121578

**Published:** 2023-06-07

**Authors:** Hera Hasan, Mohammad Afzal, Javier S. Castresana, Mehdi H. Shahi

**Affiliations:** 1Interdisciplinary Brain Research Centre, Faculty of Medicine, Aligarh Muslim University, Aligarh 202002, India; herahasanofficial786@gmail.com; 2Department of Zoology, Faculty of Life Sciences, Aligarh Muslim University, Aligarh 202002, India; ma.afzal1235@gmail.com; 3Department of Biochemistry and Genetics, University of Navarra School of Sciences, 31008 Pamplona, Spain; jscastresana@unav.es

**Keywords:** glioblastoma, miRNA, temozolomide, epigenetics, DNA methylation

## Abstract

Glioblastoma is the most aggressive form of brain tumor originating from glial cells with a maximum life expectancy of 14.6 months. Despite the establishment of multiple promising therapies, the clinical outcome of glioblastoma patients is abysmal. Drug resistance has been identified as a major factor contributing to the failure of current multimodal therapy. Epigenetic modification, especially DNA methylation has been identified as a major regulatory mechanism behind glioblastoma progression. In addition, miRNAs, a class of non-coding RNA, have been found to play a role in the regulation as well as in the diagnosis of glioblastoma. The relationship between epigenetics, drug resistance, and glioblastoma progression has been clearly demonstrated. *MGMT* hypermethylation, leading to a lack of *MGMT* expression, is associated with a cytotoxic effect of TMZ in GBM, while resistance to TMZ frequently appears in *MGMT* non-methylated GBM. In this review, we will elaborate on known miRNAs linked to glioblastoma; their distinctive oncogenic or tumor suppressor roles; and how epigenetic modification of miRNAs, particularly via methylation, leads to their upregulation or downregulation in glioblastoma. Moreover, we will try to identify those miRNAs that might be potential regulators of *MGMT* expression and their role as predictors of tumor response to temozolomide treatment. Although we do not impact clinical data and survival, we open possible experimental approaches to treat GBM, although they should be further validated with clinically oriented studies.

## 1. Introduction

### The CNS5 Classification and Glioblastoma

Glioblastoma, according to the 2021 WHO (World Health Organization) Classification of Tumors of the Central Nervous System, or CNS5 classification [1], is an *IDH* wild-type glioma. Previous classifications considered glioblastoma as primary (*IDH* wild type) and secondary (*IDH* mutant) [2]. CNS5 classification has simplified the subdivisions of adult-type diffuse gliomas, dividing them into three tumor types: astrocytoma (*IDH* mutant), oligodendroglioma (*IDH* mutant and 1p/19q codeletion), and glioblastoma (*IDH* wild type). 

Glioblastoma, according to the CNS5 classification, is defined as a diffuse astrocytic WHO grade 4 glioma, *IDH* wild type, and H3 wild type, that has one or more of the following histological or genetic features: microvascular proliferation, necrosis, *TERT* promoter mutation, *EGFR* amplification, and +7/−10 chromosome and copy-number changes at chromosomes 7 (gains and amplifications) and 10 (losses) [1,2].

Despite the availability of a plethora of aggressive treatments and conventional therapies, such as chemotherapy, radiotherapy, and surgical resection, glioblastoma (GBM) is still irremediable, obnoxiously fatal, and almost invariably leads to patient death. GBM stem cells (GSCs) constitute a subpopulation of GBM cells that contribute to the failure of conventional treatments, as they are highly tumorigenic but relatively quiescent cells, which make them resistant to conventional therapies [3]. Some idiosyncrasies of GBM have been identified, which make GBM a very particular entity, separated from the rest of gliomas. GBM exhibit central necrosis and microvascular proliferation, are highly infiltrative, and rarely display extracranial metastases; they are exceptionally invasive, and their rate of migration is so high that they are spread far away from their origin and even extend across the contralateral brain hemisphere, making complete surgical resection of GBM achingly complicated. GBM is highly vascularized and presents a high mutation rate and significant genetic instability that contributes to intra-tumor heterogeneity, which complicates therapy [4]. By understanding the biology and the cellular origin of GBM, the daunting task of developing potential novel approaches for treating this disease might be achieved.

Epigenetics is broadly defined as the study of heritable aberrations in gene expression without any change in DNA sequence [5]. Moreover, the reversible nature of epigenetic modification has engendered an endeavor to develop more progressive novel therapeutic approaches aiming to combat GBM [6]. Epigenetic modification plays a critical role in the transformation of normal to malignant cells via a complex interplay with genetic alterations, impacting critical cellular processes involved in the progression of glioma, such as DNA repair, apoptosis, and cell invasion and proliferation, which profoundly contribute to the catastrophic disruption of normal cells, transforming them to high-grade glioma cells [7]. DNA methylation and histone modification are the two classic most important types of epigenetic modification. However, recently, the role of non-coding RNA, such as miRNAs, and their epigenetic modifications are on the road to discovering more reliable and novel prognostic and predictive GBM biomarkers [8].

MicroRNAs (miRNAs) are small non-coding endogenous RNAs, typically 21 to 23 nucleotides long. They exert their regulatory effect post-transcriptionally by regulating a large number of genes through silencing the expression of specific mRNAs, a process called RNA interference (RNAi). Additionally, miRNAs can regulate the expression of a plethora of genes that play a key role in cancer. For example, EZH2, a chromatin modifier, has been reported to be regulated directly by miR-205, mRNA-101, and miR-26a [9]. miRNAs play crucial roles in GBM progression either by acting as oncogenic miRNAs via silencing tumor suppressor genes or by acting as tumor suppressors. Moreover, several miRNAs involved in GBM development have been observed to be methylation sensitive, which can be a potential target to develop mRNA-based therapies for GBM [8].

## 2. DNA Methylation

DNA methylation is a type of epigenetic modification that plays a key role in the regulation of chromatin structure, genome imprinting, and gene expression [10]. Most studies based on DNA methylation as an epigenetic mark predominantly focuses on methylation of a cytosine residue followed by a guanine, a reaction in which a methyl group is covalently transferred to the 5′-position of cytosine, forming 5-methyl cytosine (5mC), which occurs via an enzyme called DNA methyl transferase (DNMT) [7]. DNMT family consists of 5 members: DNMT1, DNMT2, DNMT3a, DNMT3b, and DNMT3L [7]. Among them, DNMT1 specifically targets hemi-methylated DNA, while DNMT3a and DNMT3b are involved in the de novo methylation of unmethylated substrates [6,7].

DNA methylation usually inhibits the transcription of eukaryotic genes, particularly when it occurs in their promoters. DNMTs transfer methyl residues from SAM (S-adenosyl methionine) to DNA, thus preventing the binding of transcription factors to the promoter region of target genes. DNA methylation usually targets CpG islands, which are clusters of CpG dinucleotides that are found in the promoter regions in nearly half of all human genes [7]. CpG islands are genomic regions of 200 base pairs with a 50% GC content of total nucleotides and a CpG ratio > 0.6 [10]. CpG islands are hypomethylated under standard physiological conditions [6], which correlates with active gene expression, whereas CpG islands of tumor cells have been found to be hypermethylated in DNA repair genes and tumor suppressor genes involved in cell proliferation and progression, which results in transcriptional silencing of these genes [6].

### 2.1. DNA Methylation in GBM

Cancer-specific DNA methylation is characterized by the global loss of methylation accompanied by gene promoter hypermethylation. Specific CpG hypermethylation of tumor suppressor gene promoters resulting in transcriptional silencing is the most widely studied epigenetic aberration in GBM [7,11]. While hypomethylation takes place in the repetitive region of the DNA that might be responsible for genomic instability contributing to tumor growth [7], hypermethylation of the *MGMT* (O^6^-methylguanine-DNA methyltransferase) promoter leads to *MGMT* epigenetic silencing in about 40% of GBM, which correlates with a better response to temozolomide (TMZ) treatment with an increase in median survival time [7,10].

However, the link of *MGMT* hypermethylation with a positive response to TMZ is not always guaranteed, as not all the cases of GBM with methylated *MGMT* exhibit a promising response to TMZ [7]; additionally, there are cases of better response to TMZ treatment in a patient with unmethylated *MGMT* promoter. This led us to contemplate miRNAs as a different regulatory mechanism that might regulate *MGMT* expression [11]. A negative correlation between miRNA and *MGMT* expression has been reported [12], something we will discuss later separately in this paper.

Besides this, there is another common epigenetic marker of glioma, 5hmC (DNA-5-hydroxymethylcytosine), which results from oxidation of 5mC, a reaction catalyzed via TET protein, an event of alteration of methyl status of DNA, reported in several patients with GBM. 5hmC has been reported to be negatively correlated with cell proliferation and grades of tumors. Hence, 5hmC can be considered a marker of good prognosis in GBM [7,13]. 

### 2.2. DNA Methylation: Its Role in Regulation of Metabolism in GBM

DNA methylation has been reported to be involved in the modification of genes related to glycolysis in GBM. DNA-methylating enzymes exert their regulatory effect on metabolic target genes, either via methylating DNA in introns or in the promoter region, thus silencing the expression of these genes.

### 2.3. PKM2

This enzyme controls the synthesis of pyruvate, the last metabolite of the glycolytic pathway. Hypomethylation at intron 1 of the *PKM* gene has been observed to exhibit a positive correlation with GBM. Additionally, *PKM2* is under the regulation of miR-7, miR-326, and Let miR-7. Hence, *PKM2* might be a useful target for several cancer-suppressing drugs [14].

### 2.4. LDHA

*LDHA* gene codes for lactate dehydrogenase, a glycolytic enzyme. Epigenetic silencing of this gene via hypermethylation of their promoter region has been documented in *IDH*-mutant GBM, according to classifications previous to CNS5 [14,15]. However, even silencing *LDHA* by siRNA inhibited proliferation, induced apoptosis, and increased chemosensitivity to temozolomide in GBM (CNS5 definition) cells [16].

### 2.5. HK2

*HK2* gene codes for hexokinase that catalyzes the rate-limiting step of glycolysis: phosphorylating glucose to glucose-6-phosphate. Aberration in the methylation status of this gene either via hypermethylation or hypomethylation of CpG islands within intron 1 has been reported in GBM cell lines or tumors [14]. Predominantly, hypomethylation has been found to be a frequent event in GBM compared to normal cells [14,17]. Several studies authenticate the *HK2* overexpression contribution to tumor growth with a drug-resistant phenotype [17]. Additionally, *HK2* has been observed to be silenced epigenetically in normal cells, while its overexpression in GBM is a consequence of hypomethylation of its promoter region [17]. Decitabine (a cytosine analog) is a potent DNMT inhibitor that recovers *HK2* expression in GBM, something that may prolong patients’ mean survival time [14].

Hence, these reports suggest the significance of DNA methylation in determining the possible outcome of GBM and provide insights into the highly intertwined relationship between metabolic and epigenetic regulation and their combined contribution in deciding the fate of tumor progression in GBM [17]. Thus, by understanding the complex interplays between these regulatory bodies, it will be possible to develop novel therapies which might remarkably enhance the efficacy of treatment in GBM.

## 3. miRNA

MicroRNAs (miRNAs/miRs) are small non-coding RNAs of 19–22 nucleotides long [12,18] profoundly involved in post-transcriptional regulation of expression of a plenitude of genes either via sequence-specific repression of mRNA [12] or mRNA degradation. The first miRNA was discovered in Caenorhabditis elegans in 1993 [19]. miRNAs play a pivotal role as modulators of cellular homeostasis and regulate several major cellular processes, such as proliferation, migration, cell cycle progression, and apoptosis [19,20]. Dysregulation of miRNA unequivocally has been associated with a wide range of clinical conditions, such as cancer, neurodegenerative diseases, and cardiovascular diseases. Additionally, aberrant miRNA expression has been reported to play a key predictive factor in the progression, development, and sustainment of GBM [21], either by miRNAs acting as oncogenes, silencing tumor-suppressing genes, or by themselves as tumor suppressors [22]. 

### 3.1. miRNA Biogenesis

miRNA biogenesis can progress through two pathways: canonical and non-canonical. Among them, the canonical pathway is the most important one related to miRNA biogenesis and maturation [23]. The first step of the canonical pathway concerns the synthesis of pri-miRNA from the genome by the enzyme RNA polymerase III/II followed by its subsequent cleavage into pre-miRNA through a microprocessor complex consisting of Drosha-DGCR8 complex [19,23,24]. Drosha is a nuclear RNase III enzyme that cleaves hairpin loop sequences in pri-miRNA [24], resulting in the production of ~70 nucleotides (nt) pre-miRNA bearing a 2 nt 3′ overhang [23]. After the completion of these initial steps in the nucleus, pre-miRNA is shuttled out of the nucleus into the cytoplasm by a nuclear transporter protein Ran/GTP/Exportin-5 (XPO5) for final processing via Dicer enzyme and TARBP2 [25,26]. Dicer is an RNAase III endonuclease that cleaves a pre-miRNA into a mature miRNA duplex [25]. The final step that marks the end of miRNA biogenesis is a load of either the 3p or 5p strand of the mature miRNA duplex, depending on their thermodynamic stability, into the Argonaute (AGO) protein, making functional miRISC (miRNA-induced silencing complex) [23] recognize the 3′-UTR of a target gene and result in translational suppression or degradation of mRNA via a phenomenon called RNA interference [20]. Approximately, 50% of all protein-coding human genes are regulated by miRNAs. Therefore, any alteration in their miRNA expression might result in clinical diseases, such as cancer and others [27].

### 3.2. miRNA in GBM

GBM is characterized by aberrations in miRNA expression. miRNAs either act as oncogenes or tumor suppressors and exert a major impact on oncogenic processes involving gliomagenesis, such as regulation of angiogenesis, metabolic pathways, and associated enzymes, or by regulating GSC (glioma stem cell) differentiation in GBM. miRNAs can regulate around 3% of all genes of glioma tumors and 30% of coding genes. Interestingly, one miRNA can control the expression of 100 mRNA associated with GBM [13]. miRNAs target metabolic reprogramming, a critical hallmark of GBM [28], which results in increased aerobic glycolysis in GBM compared to normal brains. Aerobic glycolysis is controlled by oncogenic signaling pathways and tumor suppressor genes; aberration in any of these genes may alter the expression of metabolic enzymes and the activity of metabolic transporters [28]. Expression of these glycolytic regulators and metabolic genes is targeted by miRNAs, such as miR-144, miR-155, miR-34a, and others [14]. Some miRNAs also act as epigenetic regulators of GSC in GBM and may contribute to GBM heterogeneity during tumor formation. Some examples are miR-451 and miR-1275, which impede cell proliferation in GSC, while miR-137 enhances proliferation and inhibits apoptosis. miRNAs also contribute significantly to the regulation of angiogenesis, another significant hallmark of GBM, by inducing molecules, such as cytokines, metalloproteinase, and growth factors, such as VEGF or EGFR [20]. Thus, several potential drugs can be developed, and valuable therapeutic targets can be determined for GBM treatment if we do a meticulous analysis of miRNA expression patterns. In this review article, we will make a compendium of miRNAs known to be upregulated or downregulated in the GBM (Table 1 and Table 2). Additionally, we will present their different roles in gliomagenesis.

### 3.3. Upregulated miRNA

#### miR-21

miRNA-21 was the first one reported as an oncogenic miRNA with an anti-apoptotic activity significantly contributing to the progression of GBM. It is highly upregulated in most tumors, particularly in GBM [21]. Thus, by downregulating miR-21 with subsequent caspase activation, the miR-21 anti-apoptotic effect can be reversed, which will ultimately aid in enhancing overall survival in GBM patients [6]. miR-21 is a key inhibitor of PTEN and p53, and an activator of EGFR, Cyclin D1, and AKT-2 [29]. Other targets of miR-21 are proteoglycan, SPOCK1, and transcription regulators, such as RB1CC1 [7].

miR-21 enhances EGFR expression by repressing PPAR alpha and VHL, with subsequent activation of beta-catenin and AP-1. Therefore, the silencing of miR-21 expression will decrease the oncogenic activity of EGFR, BCL2, and cyclin D, and it will lead to the upregulation of tumor suppressor proteins, such as Bax, p21, TGFBR2, or p53 [30]. Apart from having pro-proliferative activity, miR-21 also promotes tumor invasion and migration [31], as it induces tumor invasion by targeting regulators of matrix metalloproteinase, such as RECK (reversion inducing cysteine-rich protein with Kazal motifs) and TIMP3 (tissue inhibitor of metalloproteinase) [24,30]. Finally, it has been reported that miR-21 silencing can be used as a therapy in the treatment of TMZ-resistant GBM patients [30].

### 3.4. miR-10b and miR-10a

These are oncogenic miRNAs commonly overexpressed in GBM [29]. miR-10a exhibits chemoresistance [31], while miR-10b significantly promotes GBM cell proliferation, invasion, migration, and EMT (epithelial–mesenchymal transition) [18], and it also imparts an oncogenic effect on GSC cells [31]. miR-10b expression levels correlate with clinical and WHO tumor grades [32]. miR-10b targets are cell cycle inhibitors [7], such as CDKN1A and CDKN2A, BIM, BCL2, TEAP2C, and PTEN, an antagonist of the PI3K pathway [14,18,31]. It promotes GBM cell invasion by enhancing RhoC (Ras homolog gene family member C) and uPAR (Urokinase-type plasminogen activator receptor) expression via modulation of HOXD10 expression [18]. Thus, miR-10b inhibition along with the co-targeting of PTEN and activation of p53 will lead to the induction of apoptosis as well as to cell cycle arrest and suppression of GBM cell invasion and migration [18,31].

### 3.5. miR-10b-miR-21 

miR-10b and miR-21 complex co-inhibition are reported to be associated with cell cycle arrest with a considerable reduction in GBM cell migration [33].

#### 3.5.1. miR-10b and miR-222 

miR-10b and miR-222 together contribute to oncogenic activity in GBM and are associated with poor survival [34]. Synergistically, they promote tumor progression and cell proliferation by targeting PTEN, which activates the p53 antitumor signaling pathway by suppressing MDM2, a key inhibitor of p53 [18,34]. They also regulate apoptosis in a p53/PTEN-independent manner directly by modulating the expression of BIM, an apoptotic initiation factor [34]. Thus, miR-10b and miR-222 can be potential therapeutic targets for the treatment of GBM [34].

#### 3.5.2. miR-9

miR-9 overexpression in GBM with poor overall prognosis has been widely reported [14,29,35]. It promotes tumor cell proliferation, migration, and inflammation; it also indirectly regulates KRAS via targeting NF1, a KRAS inhibitor [14,35]. Overexpressed miR-9 has been observed to be positively correlated with GSC differentiation, thereby conferring chemoresistance to GBM [35]. miR-9 also induces TMZ resistance in CD133+ cells; therefore, miR-9 upregulation is an indicator of poor survival [35,36]. Additionally, miR-9, when upregulated, downregulates tumor suppressor gene *PTCH1*, which results in the activation of the SHH signaling pathway, thus reducing tumor cell death [36].

#### 3.5.3. miR-221/222

miR221/222 overexpression is associated with the increase in tumor growth and other major hallmarks of cancer, such as proliferation, migration, invasion, and angiogenesis. It has an oncogenic influence on GBM, and it correlates with poor survival [7,12,18]. miR-221/222 cluster promotes all these malignant properties by increasing MMP2, MMP3, and VEGF along with its target Akt signaling pathway [12,18]. Other targets of miR-221/222 are PTEN, TIMP3, E2F3, and PUMA [12,37]. The miR-221/222 cluster displays anti-apoptotic activities by inhibiting *PUMA*, which is a proapoptotic gene, or by co-targeting the *PTEN* gene [30,37]. They also promote tumor cell proliferation by targeting the p27Kip1 cell cycle inhibitor [7], a member of cyclin-dependent kinase inhibitors that do not let the phase transition of cells from the G1 to the S phase. Thus, by downregulating the expression of the miR-221/222 cluster, tumor proliferation and growth can be controlled [13,37].

#### 3.5.4. miR-26a

This miRNA is overexpressed in high-grade GBM, and its expression is associated with the degree of malignancy [38]. It acts as an oncogene by binding to the 3′ UTR of PTEN, leading to PTEN protein inhibition. MiR-26a overexpression in glioma cells circumvents the need for loss of heterozygosity of PTEN to promote tumor formation [7,14,29]. miR-26a expression has been found to be directly upregulated by MYC oncogene [38].

#### 3.5.5. miR-17-92 Cluster

The expression of this miRNA cluster has been reported to be exceptionally higher in GBM than in normal healthy brain tissue. It employs its oncogenic effect on high-grade glioma and its elevated expression level associated with the aggressiveness of the tumor [8]. miR-17-92 higher expression correlates with GBM [39] and with GSC regulation [24]. It promotes GSC differentiation and exhibits an anti-apoptotic effect. It targets tumor suppressor genes and cell cycle inhibitors, such as *PTEN* and *CDKN1A* [8]. Thus, inhibiting or lowering the expression level of this cluster may promote a longer survival rate in GBM patients [30].

#### 3.5.6. miR-148a

This miRNA promotes exosome-induced GBM, cell proliferation, and metastasis [40]. It has been acknowledged as one of the significant GBM-associated risky miRNAs by TCGA (The Cancer Genome Atlas) [40]. miR-148a level has been found to be excessively elevated in the serum of patients with GBM compared to normal healthy participants [40]. miR-148a exerts its oncogenic influence by directly targeting *CADM1* (cell adhesion molecule 1), which is a tumor suppressor gene that inhibits cell motility and tumor proliferation [18,40]. Thus, by decreasing CADM1 and protein levels, miR-148a increases STAT 3 pathway activity and causes glioma cell proliferation and metastasis [18,40]. 

#### 3.5.7. Other Upregulated miRNA in GBM

miR-221, miR-125, miR-182, miR-196, miR-30, miR-143, miR-494-3p, miR-96a/96b, miR-182, miR-210, miR-503, and miR-378 have all been reported to be upregulated in GBM and contribute to gliomagenesis.

miR-221 targets cell cycle inhibitors [7] and promotes proliferation as well as potentially enhancing GBM cell migration and invasiveness [39]. 

miR-125 is an oncogenic miRNA and promotes the proliferation of GBM cells by targeting the anti-apoptotic gene *BMF* [29].

miR-182 and miR196 are both elevated in GBM [29]. miR-182 enhances the aggressiveness of glioma cells by targeting USP15, TNIP1, CYTLD, and OPTN, which disrupts the negative feedback loop of NF-kB. On the other hand, miR-196 increases glioma cell proliferation and poor survival in GBM patients [29].

miR-30 and miR-486 are oncogenic miRNAs and promote angiogenesis [29]. miR-30 is overexpressed in GSC, and it enhances its tumorigenicity by silencing SOCS3 (suppressor of cytokine signaling 3) along with inducing the JAK/STAT3 pathway [41]. It also induces resistance to glioma cells against TRAIL protein as well as inhibits apoptosis by binding to the 3′UTR of caspase 3 [41]. miR-30 has been observed to be negatively correlated with the survival of glioma patients [41].

miR-143 promotes glioma cell differentiation by targeting HKII [14]. miR-451 is an oncogenic miRNA; it upregulates in GBM and contributes to tumorigenesis by targeting the LKB39-AMPK pathway [14].

MiR-495-3p promotes proliferation, migration, and invasion by targeting the P13K/AKT tumor-suppressing signaling pathway [13]. miR-96a and 96b are oncogenic miRNAs that contribute to the poor overall survival of glioma patients [13]. miR-93 promotes angiogenesis and tumor growth [13]. miR-503 is upregulated in GBM and inhibits apoptosis by targeting PACD4 [18]. miR-378 targets VEGFR2 and promotes tumor growth and angiogenesis [18,20]. miR-201 is an oncogenic miRNA, significantly elevated in the serum of glioma patients. It is associated with high-grade glioma and poor overall survival [42]. The major target of mir-201 is HIF (hypoxia-inducible factors 1 and 2). It induces cell proliferation and inhibits apoptosis [18,42]. Thus, it can be considered as a potential circulating biomarker but not preferable for early detection of glioma [42].

### 3.6. Downregulated miRNAs

#### 3.6.1. miR-31

It plays a pivotal role in tumor suppression by inhibiting invasion and migration via targeting the *RDX* gene (Radixin) [29].

#### 3.6.2. miR-124

miRNA-124 has a low expression in high-grade glioma [43]. Its overexpression lowers SNAI2 levels, which results in suppression of GSCs invasiveness and of other stem-like traits that contribute to GBM malignancy [29]. It also induces cell cycle arrest directly by targeting CDK4, CDK6, and cyclin D. Hence, mir-124 inhibits glioma cell growth [43].

#### 3.6.3. miR-34

miR-34a is downregulated in GBM, and, if expressed in tumor cells, it causes suppression of cell proliferation and invasion, controls GSC differentiation, stem-like traits, and cell cycle arrest by targeting Notch1, Notch2, CDK6, EGFR, and c-Met [13,44]. They also promote apoptosis by targeting Bcl-2 [45,46].

miR-34c-3p and miR-34c-5p present very low expression in high-grade glioma, which indicates that these miRNAs have a tumor-suppressing impact on gliomagenesis [44]. Overexpression of these miRNAs results in the suppression of tumor invasion [46]. miR-34c-3p targets Notch2 and lowers its expression, inhibiting cell proliferation and promoting S-phase arrest, along with the activation of apoptotic pathways [44]. miR-34c-5p expression correlates with a decrease in Notch1 and Notch2 levels, which results in the inhibition of cell proliferation. Targets of miR-34c-5p other than Notch1/Notch2 are CDK6 and EGFR [44].

#### 3.6.4. miR-302-367 Cluster

This miRNA targets GICs (glioma-initiating cells), which confer resistance to glioma cells against TMZ treatment, as well as inhibits the CXRC4 receptor, which, in turn, disrupts the SHH signaling pathway, thus resulting in preventing tumor progression [14]. This cluster of tumors suppressing miRNAs, when expressed in GBM cells, targets transformation-related proteins required for the maintenance of tumor stemness by suppressing the expression of reprogramming factors, such as SOX2, Myc, KLF4, and OCT3/4, and transcription factors, such as SALL2 and OLIG2. Along with this, they enhance the expression of tumor suppressor genes, such as *UCH1*, *PEA15*, and *MYBBP1A*. miR-302-367 cluster induces differentiation of glioma cells by suppressing the expression of stem-like genetic programming by inhibiting PI3K/AKT and STAT3 signaling pathways [47].

#### 3.6.5. miR-181

miR-181 is a family of four members (miR181a, miR-181b, miR-181c, and miR-181d), all of which are downregulated in GBM [21,48]. Among them, a significant reduction in expression levels of miR-181a and 181b has been observed in high-grade gliomas. These two members of miR-181 are routinely employed in distinguishing high-grade gliomas from low-grade gliomas [21]. miR-181a suppresses GSC-induced stemness and other tumorigenic effects via targeting CD133 and BMI1 stemness-related markers [21]. miR-181c expression decreases in GBM due to the absence of CTCF with concomitant epigenetic silencing by DNA methylation [9]. miR-181d forms a cluster with miR-181c on chromosome 19, and it is downregulated in GBM. miR-181d regulates the WNT signaling pathway by targeting the *CREBBP* gene [48]. Its expression level negatively correlates with IGF-1 in GBM [48,49]. IGF-1 promotes tumor progression by inhibiting miR-181d with modulating cytokines secretion, with a concomitant increase in cytokines level [49]. miR-181d, therefore, exhibits a profound correlation with IGF-1-associated cytokines. These upregulated chemokines in GBM, such as C-C chemokine receptor type 1 (CCR1) and interleukin (IL-1b) are the direct targets of 181d tumor-suppressing activity [49]. The other important targets of miR-181d for tumor suppression are Bcl-2 and KRAS [49]. Thus, through a meticulous assessment of the expression level status of IGF-1 and miR-181d in GBM tissues, the overall survival rate in a GBM patient might be determined [49].

#### 3.6.6. miR-219-5p and miR-219-1-3p

Downregulation of these miRNAs correlates with the increase in glioma cell proliferation. Their overexpression causes a reduction in tumor growth [13].

#### 3.6.7. miR-1

miR-1 has a tumor-suppressing activity in GBM cells, as it has been observed that it inhibits the proliferation and migration of GBM cells when expressed ectopically [18,21,50]. Additionally, upon expression in glioma cells, they enhance the sensitivity of GBM cells toward TMZ induce apoptosis [50].

#### 3.6.8. miR-370-3p

miR-370-3p is downregulated in both high- and low-grade glioma [12]. They suppress cell proliferation and migration by regulating the WNT signaling pathway via targeting the 3′UTR of β-catenin whose stabilization is pre-required for activation of WNT signaling [12]. Other targets of miR-370-3p are FOX01 (forkhead box 01 in humans), FOXM1, and TGFβ (Transforming growth factor β) [51]. Therefore, miR-370-3p can be used as a potential target in the development of anti-GBM therapy [51].

#### 3.6.9. miR-328

miR-328 lower expression suggests a poor overall survival rate in patients with GBM [21]. The downregulation of miR-328 in GBM tissues contributes to proliferation and tumor growth by enhancing cell division. miR-328 anti-proliferative activity can be used as a potential therapeutic target for GBM therapy [52].

#### 3.6.10. miR-375 

miR-375 is an anti-proliferative miRNA whose downregulation aids in the progression of glioma cell tumorigenesis by facilitating its cell proliferation, invasion, and migration [18,53].

#### 3.6.11. miR-137

This is a tumor-suppressing miRNA that, on expression, provides protection from tumor progression by inhibiting angiogenesis via inhibition of EZH2 (enhancer of zest homology2), a key proliferation-inducing factor [18,20]. Thus, low expression of this miRNA often correlates with poor prognosis [18,20].

#### 3.6.12. miR-128

This miRNA has a prolific role as a tumor-suppressing factor. Its low expression is associated with high-grade glioma. Therefore, miR-128 can be used to distinguish between low- and high-grade glioma [21]. miR-128 induces its tumor-suppressing effect, such as anti-proliferative and anti-metastasis effects, by inhibiting tumor-associated signaling pathways, such as WNT, ERK, EGFR, IGF1R, or BCL2 [13,18,24,54]. miR-128 induces apoptosis via caspase activation [21]. It suppresses GSC’s self-renewable capacity by targeting SUZ12, E2F3, and BMI1 [24,54]. miR-128 reduces tumor cell growth by targeting PDGFRA and EGFR and controls angiogenesis by inhibiting P70S6K1 kinase [55].

#### 3.6.13. miR-7

miR-7 is downregulated in low-grade gliomas [14]. Consequently, the EGFR expression level increases, inducing upregulation of PKM2 via NF-κB activation, thus contributing to glioma tumorigenesis [13,14]. The other target of miR-7 is the AKT/PI3K signaling pathway [13]. 

**Table 1 cells-12-01578-t001:** List of upregulated miRNAs in GBM.

miRNA	Target	Expression	Function	Reference
miR-21	PTEN, p53, VH1, PPARa, TIMP3, RECK, SPOCK_1_, RB1CC1	Up	Tumor Growth (+), regulate EGFR/AKT signaling, Cell invasion (+), Cell proliferation (+), Apoptosis (−)	[7,24,29,30]
10b	p-53, CDKN1A, CDKN2A, BIM, BCL2, TEAP2C, HOXD1O, uPAR, R4OC	Up	Promotes cell cycle, Cell invasion (+)	[14,18,31]
miR-10b/222	p53/PTEN, BIM	Up	Apoptosis (+), Cell Proliferation (+)	[18,34]
miR-9	NF_1_, PTCH_1_P	Up	Cell proliferation (+), Cell migration (+), Inflammation (+), Resistance to chemotherapy (+), Apoptosis (−)	[14,35,36]
miR-221/222	PTEN, MMP2, MMP3, BEGF, PUMA, E2F3, TIMP3, P27KiP1	Up	Tumor growth (+), Apoptosis (−), Proliferation (+), Angiogenesis (+), Migration (+), Invasion (+)	[7,12,18,30,37]
miR-26a	PTEN	Up	Tumor growth (+)	[7,14,29]
miR- 148a	CADM1	Up	Cell proliferation (+), Metastasis (+)	[18,40]
miR-125	BMF	Up	Apoptosis (−)	[29]
182	USPI5, TNIP1, CMTLD	Up	GBM aggressiveness (+), Disrupt negative feedback loop of NF-KB	[29]
miR-196		Up	Cell proliferation (+), Poor survival	[29]
miR-30	SOCS3, JAK/STAT3, TRAIL Protein	Up	GSC differentiation (+), Apoptosis (−)	[41]
miR-143	HKII	Up	Cell differentiation (+)	[14]
miR-145	LKB 39- AMPK pathway	Up	Tumor growth (+)	[14]
miR-495-3p	PTEN/AKT pathway	Up	Migration (+), Proliferation (+), Invasion (+)	[13]
miR-503	PACDA	Up	Apoptosis (+)	[18]
miR-93		Up	Angiogenesis (+), Tumor growth (+)	[13]
miR-378	VEGFR2	Up	Angiogenesis (+), Tumor growth (+)	[18,20]
miR-201	HIF1, HIF2	Up	Apoptosis (−), Cell proliferation (+)	[18,42]

**Table 2 cells-12-01578-t002:** List of downregulated miRNA in GBM.

miRNA	Target	Expression	Function	Reference
miR-31	Radixin	Down	Invasion (−), Migration (−)	[29]
miR-124	SNA12, CDKA, CDK6, Cyclin D	Down	Cell cycle arrest (+), GSCs invasiveness (−)	[29,43]
miR-34a	Notch 1, Notch 2, CDK6, EGFR, C-met, BCI-2	Down	Cell proliferation (−), Invasion (−), GSCs differentiation (−), Cell cycle arrest (+)	[13,44,45,46]
miR-34c-3p	Notch 2	Down	S-phase arrest (+), Proliferation (−), Apoptosis (+)	[44]
miR-34c-5p	Notch 1, Notch 2, CDK6, EGFR	Down	Cell proliferation (−)	[44]
miR-302-367 cluster	GIC, CXRC4, PI3K/AKT pathway, STAT3 pathway, SALL2, OLIG2, SOX2, CMyC, KLF4, OCT3/4, UCH1, MYBBP1A, PEAL5	Down	Tumor growth (−), GSC stemness (−)	[14,47]
miR-181 (a) miR-181a (c) miR-181d	CDI33, BMI1, WNT signaling pathway, CCR1, IL-1b, BCI-2, K-Ra5	Down Down	GSC stemness (−) Tumor growth (+)	[21] [48,49]
(b) miR-181c	TGFBR1 TGFBR2, TGFBRAP1	Down	Cell invasion (−), Proliferation (−)	[56]
miR-219-5p miR-219-1-3p		Down	Tumor growth (−), Proliferation (−)	[13]
miR-1		Down	Sensitize GBM to TMZ, Apoptosis (+)	[50]
miR-328		Down	Proliferation (−)	[52]
miR-375		Down	Proliferation (−), Invasion (−), Migration (−)	[53]
miR-137	EZH2	Down	Angiogenesis (−), Proliferation (−)	[18,20]
miR-128	WNT, BRK, EGFR, IGF1R, BCL2, 5UZI2, BIM1, EZF3, PDGFRA	Down	Apoptosis (+), Proliferation (−), Metastasis (−), Angiogenesis (−), GSCs Renewability (−)	[13,24,54,55]
miR-7	PKM2, EGFR, AKT/PI3K pathway	Down	Tumor Growth (−)	[13,14]

## 4. miRNA and DNA Methylation: An Epigenetic Interplay in GBM

The complex epigenetic interplay between miRNA and DNA methylation has lately appeared to be quite intriguing to researchers. The monitoring of epigenetic changes during tumor progression can be quite useful to assess the efficacy of any epigenetic therapy, which might be used in combination with other established anti-tumor therapies in order to enhance their sensitivity and subdue the tumor-induced resistance against these therapies [57].

miRNA can either get modulated by epigenetic regulation or can, in turn, regulate those epigenetic modulators via feedback mechanisms. Thus, the epigenetic machinery and miRNA interaction can be considered potential targets for tumor therapies [57]. Most miRNAs are downregulated in GBM as a result of hypermethylation in the CpG island of their promoter region, a phenomenon of miRNA silencing via DNA methylation. There are plenty of miRNAs in GBM that are regulated epigenetically via DNA methylation. 

miR-29a, miR-29b, and miR-29c constitute the miRNA-29 family of tumor-suppressing miRNAs that directly target DNA methyl transferases, such as DNMT3a and DNMT3b. As a result of this interaction, DNA methylation is repressed, hence suppressing tumor progression of glioma cells [58,59]. miR-185 and miR-153 are tumor suppressors, which, when overexpressed in glioma cells inhibit DNMT1, induce hypomethylation, and inhibit tumorigenesis [7,21].

miR-211 promotes apoptosis by targeting MMP9 with concomitant activation of caspase-9/caspase-3 to inhibit tumor invasion. It has been reported to be downregulated in GBM due to epigenetic silencing via hypermethylation in its promoter region [58]. Other tumor suppressor miRNAs, such as miR-204, miR-145, miR-137, miR-124, miR-127, miR-219-1, and miR-181c, have been reported to be subjected to epigenetic silencing via DNA methylation in GBM. Among these, miR-181c epigenetic regulation in GBM has been profoundly studied and widely mentioned in the research literature.

miR-181c is a tumor suppressor miRNA that is under-expressed in GBM due to DNA-methylation-induced repression. Its expression level inversely correlates with tumor invasion and proliferation [58]. miR-181c expression is regulated via CTCF and DNA methylation. CTCF is an 11-zinc finger highly conserved nuclear protein [12,13,31], which protects miR-181c repression from DNA methylation by binding to its CpG island region of their promoter. Thus, the absence of CTCF and the gain of DNA methylation together contributes to the downregulation of miR-181c in glioma cells [58].

miR-204 targets SOX4, a stem transcription factor, and prevents cell invasion. This miRNA is downregulated in GBM via DNA methylation [58]. miR-23 is another tumor-suppressing miRNA that causes cell cycle arrest but has been found to be inactivated epigenetically [58]. miR-137 inhibits GSC differentiation, but it is under-expressed in tumor cells and GSCs cells as a consequence of hypermethylation in their promoter region [58,60]. miR-124 aberrant expression induced by epigenetic silencing is responsible for uncontrolled cell growth, as miR-124 hypermethylation prevents it from causing cell cycle arrest at Go/G1 phase [43]. miR-127 and miR-219-1 are tumor-suppressing miRNAs that have lower levels of expression in GBM than in healthy brains due to hypermethylation in their promoter region [8]. Thus, the understanding of these epigenetic networks and their interaction with different miRNAs involved in GBM might be quite useful to establish new reliable and precise approaches for the diagnosis and treatment of GBM.

## 5. miRNA and Epigenetic Modifications in TMZ Response and Drug Resistance

The main obstacle in current GBM treatment is the resistance to radiotherapy and chemotherapy (TMZ), which profoundly limits the effectiveness of these therapies. TMZ is the first-line chemotherapeutic agent, currently considered the standard therapeutic option for the treatment of GBM. However, this therapy is often susceptible to many resistance-inducing factors which limit its efficacy. miRNA and epigenetic modification greatly influence the TMZ response to GBM treatment. Thus, in order to predict and improve these therapeutic responses in GBM patients, it is necessary to elucidate the mechanisms by which miRNA and epigenetic factors control the outcome of GBM therapies, and even more specifically, it is essential to understand which factors and regulatory mechanisms influence and determine the response to TMZ in GBM patients.

### 5.1. Epigenetic Modulation and TMZ Response

MGMT is a DNA repair system considered a major contributor to TMZ resistance in GBM [61,62]. MGMT induces resistance to TMZ by removing a methyl group from O_6_-methylguanine, which results in the neutralization of TMZ-induced DNA damage, thus reducing the overall cytotoxic effect of TMZ [62]. Therefore, *MGMT* methylation status is often considered an important predictor of TMZ treatment response [6,62].

The epigenetic silencing of the *MGMT* gene via hypermethylation at the CpG islands of its promoter results in the inactivation of the *MGMT* gene, which correlates with enhanced TMZ efficacy along with better prognosis in GBM patients [6,58]. Therefore, it is essential to understand the epigenetic changes taking place during gliomagenesis in order to predict better outcomes after novel and conventional therapies.

### 5.2. miRNA and TMZ Response

Besides their particular epigenetic actions, miRNAs have been reported to play important roles in TMZ resistance [61] (Table 3). Several miRNAs have been observed to play regulatory roles in TMZ response: miR-195, miR-130a, miR-181a, miR-221, miR-21, miR-210, miR-222, and miR-10a. Apart from these, there are several other miRNAs that have been reported to be involved in MGMT regulation [62]. Upregulation of miR-370-3p, miR-603, miR-221/222, and miR-648 and downregulation of miR-181d, miR-370-3p, and miR-142-3p results in the inhibition of MGMT suppression, therefore, conferring chemoresistance to GBM cell against TMZ treatment [62]. The downregulation of miR-221/222 via antagomiRs treatment has been shown to increase the sensitivity of GBM cells to TMZ as well as promote apoptosis by restoring the p53 pathway [55,63]. We have also noted the following:
miR-21 in an oncogenic miRNA that contributes to drug resistance. Its downregulation enhances chemotherapy efficacy against human GBM cells [63]. miR-21 is often considered a potential biomarker for TMZ resistance [18]. Therefore, silencing miR-21 with simultaneous TMZ treatment can markedly enhance the apoptosis of cancer cells and, therefore, increase the median survival time of patients with TMZ-resistant GBM [18].miR-181d has also been identified as a predictor of TMZ response and patient survival [49]. It was experimentally proved that transfecting miR-181d into GBM cells caused *MGMT* expression decay, which is associated with good prognosis and overcoming of resistance. So, miR-181d positively associates with TMZ response and patient survival [18,63]. Another miRNA of the same family, miR-181c, is involved in TMZ resistance, as it is suppressed in a patient with GBM who showed a positive response to radiotherapy/TMZ treatment [64].miR-195 and miR-10a are reported to be overexpressed in GBM cells having low sensitivity to TMZ; therefore, downregulation of these miRNAs can significantly improve TMZ response and survival chance [55]. miR-124, miR-134, and miR-128 induce their antitumor activity synergistically by inhibiting GSC proliferation and promoting an effective response of radiotherapy and chemotherapy against GBM [63].miR-370-3p, a negative regulator of *MGMT*, has been reported to be highly downregulated in TMZ-resistant GBM cells. miR-370-3p suppresses *MGMT* expression in GBM cells and sensitive glioma cells to TMZ [51,65], inducing apoptosis of tumor cells [51,65]. Thus, miR-370-3p can have a potential therapeutic role in the treatment of recurring GBM if used to improve TMZ response [65].miR-128 and miR-149 overexpression sensitize glioma cells to TMZ, especially in the case of non-stem GBM cells and, therefore, contribute to better prognosis [54].miR-125b overexpression confers chemoresistance of GSCs to TMZ treatment. The combined inhibition of PI3K and miR-125b significantly enhances TMZ-induced inhibition of GSC proliferation and invasiveness [18]. miR-100 overexpression in glioma cells sensitized them to ionizing radiation by downregulating the ataxia telangiectasia mutated (*ATM*) gene [29]. miR-328 sensitizes GSC to TMZ by directly suppressing ABCG2 expression [29]. miR-218 and miR-1268a are associated with enhanced TMZ response in GBM patients [18,29]. miR-1268a is downregulated in a patient with recurrent GBM, and its overexpression promotes TMZ sensitivity to GBM cells via inhibition of translation of the *ABCCL* gene [18]. miR-299-5p enhances TMZ sensitivity to GBM cells by inhibiting cell proliferation via regulation of the ERK signaling pathway [18]. Overexpression of miR-423-5p and miR-223 promotes GBM cell survival by decreasing TMZ response [18]. miR-223 expression suppresses TMZ, inducing the inhibition of cell proliferation as well as the miR-223/PAX6 axis that further contributes to chemoresistance and decreases in TMZ response by regulating the PI3K/AKT signaling pathway.miR-318, miR-381, and miR-20a overexpression also result in increased TMZ resistance [18]. Apart from the aforementioned miRNAs, new miRNAs are continuously being discovered that can be used as potential therapeutic tools in combination with established chemotherapy and radiation therapy.

By understanding the expression profiles of these dysregulated miRNAs in TMZ-resistant glioma cells, novel therapeutic targets can be determined, which can be further used in improving the outcome of conventional therapies.

Other than these, epigenetic therapies can be used in combination with traditional therapies to improve TMZ and other cytotoxic treatment responses. Epigenetic alterations that occur during GBM and confer chemoresistance can be targeted, and their effects can be reversed via epigenetic drug treatments. For example, GBM cells treated with 5-Aza-CR cause reversal of DNA methylation, thus sensitizing GBM cells to chemotherapy [57] and overcoming TMZ resistance.

**Table 3 cells-12-01578-t003:** miRNA and epigenetic effect on TMZ response.

miRNA/Epigenetic Modulator	Expression	Effect on TMZ Response	Reference
MGMT	High	Induces TMZ resistance	[62]
miR-21	Up	Induces TMZ resistance	[18]
miR-181d	Up	Sensitizes glioma cells to TMZ	[18,63]
miR-195 and miR-10a	Up	Confers TMZ resistance to glioma cells	[55]
miR-124, miR-134, and miR-128	Up	Promotes TMZ-induced cytotoxicity of glioma cells	[63]
miR-370-3p	Down	TMZ resistance	[51,65]
miR-125b	Up	Confers chemoresistance to GSCs against TMZ	[18]
miR-128 and miR-149	Up	Sensitizes non-stem glioma cells to TMZ	[54]
miR-328	Up	Enhances TMZ response to GBM by targeting ABCG2	[29]
miR-1268a and miR-218	Up	Inhibits translation of ABCCL and enhances TMZ sensitivity	[18]
miR-299-5p	Up	Inhibits cell proliferation and enhances TMZ sensitivity by regulating ERK signaling pathway	[18]
miR-423-5p and miR-223	Up	Decreases TMZ response and promotes GBM response	[18]
miR-318, miR-381, and miR-209	Up	Increases TMZ resistance	[18]

## 6. Diagnostic and Prognostic Molecular Tools in GBM: Do miRNAs Play a Role?

Biomarkers are chemical compounds that are used to monitor the biological state of disease and to measure risks associated with it [30]. They have a high diagnostic and prognostic value, which, when applied, helps in the early detection of a disease, e.g., GBM, and makes it possible to establish an appropriate time point to elicit maximum efficacy of the proposed treatment. All these aspects culminate in early recovery and prolonged survival of the patient. Thus, biomarkers play a critical role in early diagnosis and in the prediction of possible outcomes of the disease, which may help in reducing treatment costs and in extending median survival rates [63]. 

miRNAs are regulators of the pathways that play crucial roles in GBM invasion and progression. Their expression predicts the efficacy of conventional therapies that are routinely used in GBM treatment [30]. Most miRNAs have already been reported to be dysregulated in GBM so far. Therefore, miRNAs are currently being considered as potential diagnostic and prognostic biomarkers of gliomas [66]. Several studies have validated the potential roles of circulating miRNAs, particularly found in body fluids, such as CSF, plasma, and serum, in GBM diagnosis. We present the following list of promising miRNA biomarkers for high-grade glioma, i.e., GBM (Table 4):miR-21 is a potential biomarker of GBM with 90% sensitivity and 100% specificity [63]. It has been observed to have low expression in the post-operation serum of GBM patients, suggesting its potential as a serum-derived miRNA biomarker in GBM [38]. High levels of miR-21 have been reported in the plasma of GBM patients, and these levels get lower once the tumor is removed [21]. miR-21 might be used to discriminate between different WHO grades as well as to predict overall survival time in GBM patients [63].miR-26a and miR-21 are both circulatory miRNAs that are upregulated in GBM, and their serum expression levels have been observed to be reduced after surgery [38], suggesting their importance as candidate serum-based biomarkers in the diagnosis of GBM as well as in monitoring disease progression [38]. Additionally, reduced post-operative serum level of miR-26a also indicates the humoral origin of miR-26a.miR-10b is upregulated in GBM, and its overexpression promotes GBM progression and correlates with poor prognosis [63]. Its expression level positively correlates with WHO grades of gliomas as well as with tumor invasiveness [21]. Therefore, miR-10b might be used as a biomarker to evaluate glioma invasiveness and, subsequently, in the sub-classification of different tumor grades. Additionally, the combined assessment of miR-10b and miR-21 in the serum of GBM patients can aid in predicting the therapeutic effect of bevacizumab (BVZ) because miR-10b and 21 serum levels have been reported to be very high in GBM patients and associated with increased tumor diameter in BVZ treated patients. miR-328 is downregulated in GBM and acts as a tumor suppressor. The low expression level of miR-328 correlates with poor survival rate, thus it might be used as a candidate prognostic biomarker in GBM [52]. High plasma levels of miR-21 and low plasma levels of miR-128 and miR-342-3p act as candidate biomarkers in distinguishing GBM patients from healthy individuals with remarkably high sensitivity and specificity [67]. miR-342-3p expression is reduced in the plasma of glioma patients, and it is increased after surgery or chemotherapy. Therefore, miR-342-3p might be a candidate biomarker for the diagnosis and discrimination of glioma [67]. miR-320a is a tumor suppressor miRNA, and its suppression correlates with excessive cell proliferation, invasion, and tumor growth [31]. Therefore, it might be used as a prognostic biomarker [31]. miR-146b and miR-4492 can be useful as novel biomarkers in predicting and monitoring GBM progression [31]. miR-146b is an oncogenic miRNA, and its major target is TRAF6. Downregulation of miR-146b and upregulation of TRAF6 correlate with inhibition of cell proliferation as well as apoptosis of tumor cells due to a decrease in Ki-67 expression. Hence, miR-146b might be suggested as a candidate biomarker for understanding GBM prognosis as well as in discriminating different grades of glioma [31].miR-29 plasma level serves as a potential biomarker to indicate malignancy and glioma progression from grades I-II to grades III-IV [68]. miR-454-3p serum expression levels have been found markedly increased in GBM patients, and its upregulation correlates with poor prognosis. Therefore, it can be used as a candidate prognostic biomarker [68].Sometimes, single miRNA profiling is not sufficient enough to predict glioma outcomes. In such cases, profiles of several miRNAs are suggested. Seven miRNAs, including miR-15b, miR-23a, miR-133a, miR-150, miR-197, miR-497, and miR-548b-5p, are all downregulated in grades II-IV glioma patients, and the combined expression profiling of these miRNAs might be taken as a candidate biomarker in the prediction of GBM malignancy [68]. miR-181 is widely reported to be downregulated in GBM, especially in the early stages of this tumor [68]. Therefore, miR-181 might be used as a candidate biomarker for early prediction as well as in the identification of tumor grade. miR-181b and miR-181c act as predictive biomarkers of TMZ response in GBM [68] and may also help in choosing patients who are suitable for adjuvant therapy [30].miR-221/222 is found to be significantly upregulated in plasma samples of glioma patients [30,69], and its overexpression contributes to poor prognosis and low survival rates [69]. The study conducted by Zhang R et al. has confirmed that miR-221 and miR-222 might be used as potential diagnostic and prognostic biomarkers [69].

**Table 4 cells-12-01578-t004:** Dysregulated miRNA might be considered diagnostic or prognostic candidates in GBM.

Over-Expressed	Under-Expressed	Source Reference
miR-21		Serum [38] Plasma [63]
miR-26a and miR-21		Serum [38]
miR-21	miR-128 and miR-342-3p	Plasma [67]
miR-10b and miR-21		Serum [56]
miR-320a		
miR-146b		[31]
miR-454-3p		Serum [67]
miR-29		Plasma [67]
	miR-23a, miR-133a, miR-150, miR-197, miR-497, and miR-548b-5p	[67]
miR-221/222		Plasma [30,69]
	miR-181	[68]

## 7. GBM Therapy

As we have discussed in the previous section of this review, we can consider miRNAs as promising therapeutic targets on which potential GBM therapies can be developed. Recently, efforts have been made to characterize tumor-suppressing and oncogenic miRNAs involved in GBM that can be used as potential targets; some of them have even shown considerable efficacy [67]. The primary goal of miRNA-based therapies is to identify dysregulated miRNA and either inhibit those oncogenic miRNAs or replace the tumor suppressor miRNAs [67]. In addition to miRNA-based therapies, epigenetic therapies are also gaining lots of approval in the treatment of GBM. Many epigenetic drugs line azacytidine and decitabine have already been approved by the FDA for the treatment of different cancers. Apart from this, adjuvant therapies have also gained attention in recent years and have been reported to prolong overall survival [3]. Other than these therapies, molecular therapies based on alterations in miRNA genetic profiling has made a great advancement in recent years [70]. Studies have reported a significant efficacy of molecular targeted therapy in glioma treatment either in cases when they are applied alone or when applied in combination with cytotoxic chemotherapy [67]. Nevertheless, miRNA-based replacement therapy and oligonucleotide therapy are still candidate approaches but have not yet been used for therapy against GBM.

### 7.1. miRNA Based Glioma Therapy

Several oncogenic and tumor-suppressing miRNAs that have a deep impact on tumor progression and their aggressiveness have been identified, and these miRNA signatures have led to the development of miRNA-based therapies, such as miRNA replacement therapy (Figure 1), oligonucleotide therapy (Figure 2), etc. Amongst several oncogenic miRNAs, miR-21 and miR-10b [67] have been used as potential targets of oligonucleotide therapy, and several studies have demonstrated their pre-clinical efficacy. Apart from this, several tumor-suppressive miRNAs have also been identified as potential therapeutic targets, such as miR-34a, miR-128, and miR-182 [67]. Among them, miR-34 has received exceptional attention and has been reported to be entered into a phase-I trial, but it is reported to have inflammatory side effects. Despite that, it is still under trial [67].

#### 7.1.1. miRNA-Based Replacement Therapy

The principle behind the miRNA-based replacement therapy (Figure 1) is to restore or increase the activity of tumor-suppressing miRNAs by delivering exogenous. miRNA mimics are chemically synthesized miRNAs of 17-26 nts that have the same sequence as endogenous miRNAs [55,71] and can be used against GBM cells in order to inhibit cell proliferation. Several miRNA mimics are under preclinical trials and have been reported to inhibit GBM growth:

##### miR-34a 

It is a tumor-suppressing miRNA that is downregulated in GBM. miR-34a, via miR-34a mimic, restores miR-34a anti-tumor activity and leads to its overexpression, which induces cell death by targeting p53, BCL-2, KRAS, and MAPK [55,71]. 

##### miRNA-7

miRNA-7 is a tumor suppressor whose reduced expression level in GBM correlates with a high level of EGFR, which leads to uncontrolled cell proliferation. Thus, miR-7-based replacement therapy can be effective in controlling the recurrence of GBM. A study conducted on miR-7-based replacement therapy by Alamdari et al. [71] confirms that transfection of miR-7 mimic into human U373-MG GBM cell shows significant suppression in EGFR mRNA and protein level, as well as the inhibition of cell growth.

#### 7.1.2. Oligonucleotide Therapy 

This is one of the rapidly emerging anti-cancer drug therapies that have been approved by the FDA [72]. This therapy (Figure 2) is based on the Watson–Crick base pairing targeting mRNA resulting in gene silencing or alteration in the splicing pattern [72]. Oligonucleotide therapy includes ASOs, siRNA, miRNA, DNA enzymes, and aptamers, and it acts either via splicing modulation, gene correction, or translation termination [72]. Oligonucleotide therapy has been validated as a potential therapeutic approach because of its high specificity and sensitivity as well as its low toxicity [37]. Oligonucleotide therapy based on miRNA inhibition therapy can be proved as a potential therapy in the treatment of GBM. For example, miR-21, which is an oncogenic miRNA, can be inhibited via antagomiRs [37]. Injecting miR-21 antisense oligonucleotide in complex with amphiphilic R3V6 peptide in glioma cells causes inhibition of miR-21, resulting in suppression of tumor growth [37]. Another such example is miR-10b, which is also an oncogenic miRNA whose overexpression correlates with cell invasion and anti-apoptosis. miR-10b can be inhibited by PS2-O-MOE anti-miR-10b oligonucleotide, thus slowing down tumor growth [37]. These studies suggest that oligonucleotide therapy holds a promising future for GBM treatment.

### 7.2. Epigenetic Therapy

Epigenetics is one of the major mechanisms that contributes to and governs gliomagenesis. The epigenetic modulators that lead to epigenetic alterations can be potential therapeutic targets to try to reverse these epigenetic effects [73]. DNMT, enzymes, and genes, such as EZH2 and BMI1, are epigenetic modifiers [13]. The altered expression of enzymes has been identified as a pivotal epigenetic drug target for GBM treatment [73], and it is currently under preclinical and clinical trials [73].

#### 7.2.1. DNMT Inhibitors

DNA methyltransferases are responsible for de novo DNA methylation at CpG islands and impart resistance of glioma cells to chemotherapeutic drugs, such as TMZ [73]. DNMTs are also considered epigenetic drug targets. Epigenetic drugs may inhibit DNMTs, which, then, would induce hypomethylation in gene promoter regions and expression of tumor-suppressing genes [13]. Some FDA-approved DNMT inhibitors are azacytidine and decitabine, but their effectiveness in the treatment of solid tumors still has to be approved [13]. A study conducted by Ghasemi et al. [8] demonstrated the effect of 5-aza-dc on methylation and expression levels of miR-219-1. They reported that 5-aza-dc on demethylated DNA resulted in increased expression of miR-219-1 by several folds, which, in turn, decreased cyclin A2 levels and thus prevented cell proliferation.

#### 7.2.2. Histone Deacetylase Inhibitors (HDACIs)

HDACIs have shown tremendous potential as anti-cancerous drugs because they can activate genes that are silenced in GBM cells. Several research studies are under clinical trials [73]. HDAC inhibitors induce their anti-tumor effect by inducing cell cycle arrest in the G1 and G2 phases of the cycle [13,73], inhibiting angiogenesis and metastasis, as well as promoting apoptosis of glioma cells [13]. HDAC inhibitors that are reported to induce radio sensitization are valproic acid (VPA) and entinostat, thus making GBM more susceptible to radio therapy [73]. These studies suggest that thorough investigation should be carried out on these epigenetic drugs to elicit their maximum benefits as cancer therapeutic agents.

### 7.3. Molecular Target Therapy

Molecular-based therapies use small molecule inhibitors or monoclonal antibodies to inhibit growth factor pathways, angiogenesis pathways, and intracellular signaling pathways, such as PI3K/AKT/mTOR, which are involved in GBM progression [3,70]. The mechanism behind this therapy employs drugs to block signaling pathways that promote cell growth [70]. Monoclonal antibodies, such as imatinib, inhibit PDGF, a promoter of tumor growth [3], and other pathways, such as RTKs, unfortunately, lack efficacy [3]. Similarly, gefitinib and erlotinib, which are anti-EGFR drugs, are under clinical trials, but they have not shown any promising outcomes. Contrary to these drugs, bevacizumab, a monoclonal antibody that targets VEGF, has shown promising therapeutic efficacies and has been observed to be effective in promoting progression-free survival during clinical trials [3]. The combinational therapy of bevacizumab with radiotherapy and TMZ is currently under clinical trial [3]. Thus, overall molecular-based therapeutic agents are still in trial stages I/II, and they lack the desirable efficacy due to high toxicity [3]. 

### 7.4. Adjuvant Therapy

Chemotherapies, such as TMZ and radiotherapy, are the conventional anti-cancer therapies that are integrally involved in the clinical management system of many cancers including GBM [74]. However, the combination of radiotherapy and TMZ after surgical resection has been shown to increase median survival time from 12.1 months to 14.6 months in a remarkable trial conducted in 2005 by Stupp et.al. [74]. This study establishes adjuvant therapy to be a promising anti-cancer therapy for the treatment of GBM with high efficacy. However, the study fails to prove the efficacy of adjuvant therapy in p-GBM. Also, this therapy encounters tumor cell resistance to TMZ conferred by unmethylated *MGMT*, thereby decreasing the responsiveness of the therapy [3].

## 8. Challenges and Limitations

Despite the establishment and development of many anti-cancer therapies, such as chemotherapy, radiotherapy and adjuvant therapies, epigenetic therapies, and miRNA-based therapy, the prognosis of GBM is still poor. The main reason behind this is tumor heterogeneity, which contributes to drug resistance [70] and tumor cell infiltration, making surgical resection impossible. Other than high heterogeneity, the existence of biological barriers, such as the blood–brain barrier, makes it difficult to deliver miRNA across it [71]. Another challenge to miRNA delivery is the undesirable toxicities due to the activation of the innate immune system [71]. Besides this, there are other therapies, such as oligonucleotide therapies. They also face challenges, and their efficacy is limited by the presence of a blood–brain barrier as molar quantities of oligonucleotides are needed, which makes it difficult to penetrate the blood–brain barrier, thus reducing the therapeutic effect [37]. Molecularly targeted therapies are still under clinical trials and have not shown much desirable efficacy. Additionally, tumor heterogeneity confers chemoresistance to drugs associated with molecular targeted therapy, thus limiting its overall anti-cancer effect [70].

Our study is limited by the fact that no application to the clinic can be extracted from it. Rather, on the contrary, the impact of this work is on the line of the presentation and description of the important miRNAs and their epigenetic influence in GBM, linked to TMZ treatment and to *MGMT* expression, mainly (Figure 3). We consider this to be the first approach to the understanding of miRNA in GBM pathogenesis. A clinical approach would have required our knowledge of the clinical data and survival analyses in all articles reviewed, something that we assume to be very difficult or impossible to do. Our work concentrates much more on describing the importance of miRNA in GBM and their epigenetic effects without entering clinical aspects. From here, we can further jump, with different studies, to a clinically oriented approach in order to, e.g., also include the different epigenetic regulation pre- and post-therapy in GBM, with TMZ and/or any other treatment that might be subjected to testing.

## 9. Perspective and Conclusions

miRNAs and epigenetic alterations in gliomagenesis can provide an understanding of tumor progression, and thus, both can be used as potential diagnostic and prognostic tools and may help in taking clinical decisions. Circulating miRNAs can be used as potential biomarkers for early detection of GBM, which may result in increasing the overall life expectancy of glioma patients. Besides this, epigenetic modulators are gaining attention exceptionally, and they can be used as potential epigenetic drug targets to reverse epigenetic effects that contribute to tumorigenesis. Epigenetic-based therapies may also render the susceptibility of tumor cells to TMZ.

To improve the efficacies of various anti-tumor therapies, efforts should be made to identify prognostic markers for intra-tumor heterogeneity, which will aid in predicting and in the improvement of therapeutic results. To maximize the efficacy of oligonucleotide therapy, nanoparticles as a miRNA delivery system might be employed to test whether this strategy might constitute a promising therapy in the treatment of GBM.

Anyhow, we assume our work is just a small contribution to the understanding of miRNA participation in gliomagenesis, specifically performed at the level of describing the relevant participating miRNA, and the role they present in the TMZ treatment against GBM, principally influenced, to our knowledge, by *MGMT* promoter methylation. This work is an open door for others to test these molecular mechanisms in relation to clinical data and to various different experimental treatments in the following years.

## Figures and Tables

**Figure 1 cells-12-01578-f001:**
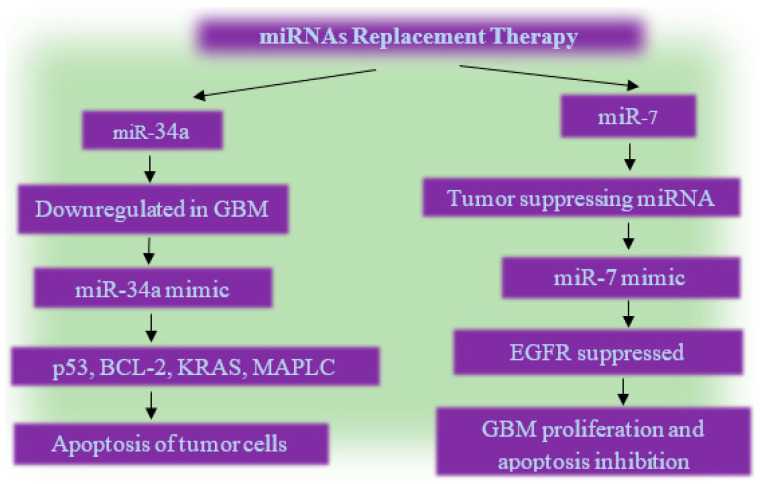
A flow chart of miRNAs replacement therapy, as a candidate experimental approach, but not yet used for therapy against GBM.

**Figure 2 cells-12-01578-f002:**
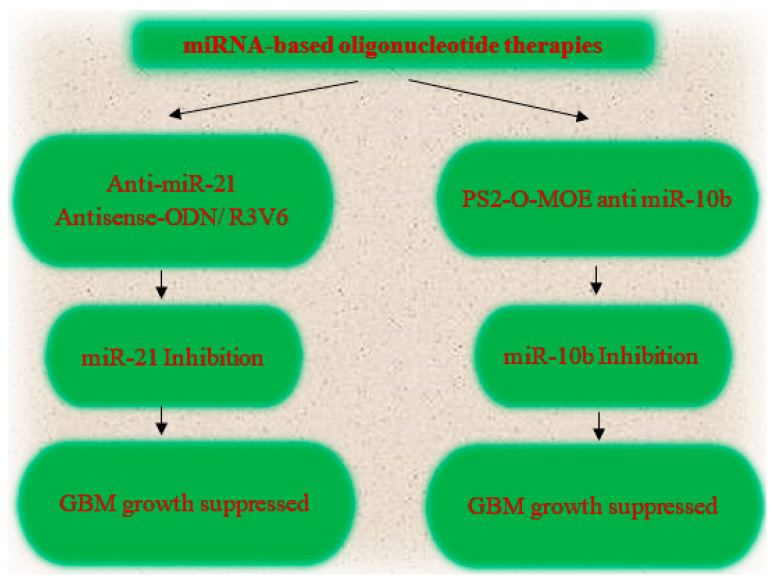
A flowchart of miRNA-based oligonucleotide therapy as a candidate experimental approach but not yet used for therapy against GBM.

**Figure 3 cells-12-01578-f003:**
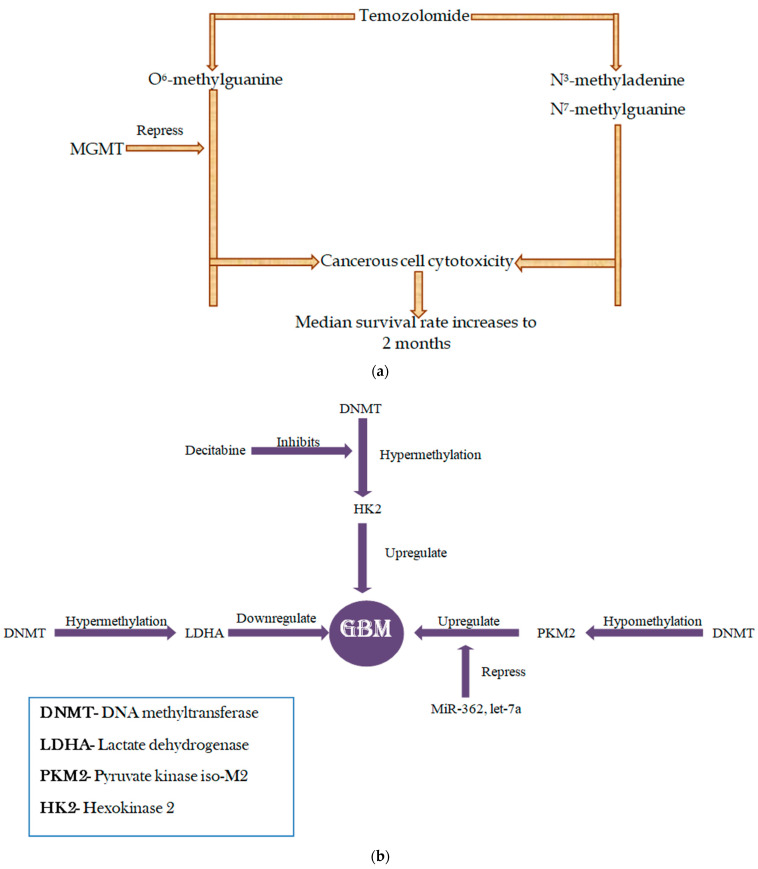
Overview of MGMT-TMZ relationship and possible experimental treatments against GBM: (**a**) Response to TMZ in GBM and role of epigenetic modifications: TMZ is a chemotherapeutic drug that kills tumor cells by methylating DNA at O6 and N7 position of guanine and at N3 position of adenine. However, TMZ efficacy is often limited by the DNA repair enzyme MGMT. Hypermethylation at the promoter region of *MGMT* can restore TMZ cytotoxic effect. (**b**) MiRNA and epigenetic regulation of glycolytic enzymes that are critical for GBM pathophysiology; PKM2, HK2, and LDHA are the glycolytic enzymes that are aberrantly expressed in glioma patients and strongly influenced by miRNA and epigenetic factors. HK2 hypermethylation exacerbates GBM progression, an effect that can be suppressed by inhibiting DNMT (DNA methyltransferases) with decitabine. On the other hand, hypermethylation of LDHA impedes GBM growth. PKM2 is regulated by both miRNAs as well as epigenetic factors. When hypomethylated by DNMT, PKM2 contributes to the growth and aggressiveness of GBM. Contrary to it, when PKM2 is targeted by miR-7, let-miR-7 and miR-326 repression of glioma is produced. (**c**) A possible interplay of non-clinical yet experimental possibilities of treatment against GBM.

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
