# Peer review of "A Comprehensive Review of miRNAs and Their Epigenetic Effects in Glioblastoma"

_cells, 2023, doi:10.3390/cells12121578_

Round 1
Reviewer 1 Report
The article "A Comprehensive Review of miRNAs and Their Epigenetic Effects in Glioblastoma" by Hasan et al. discusses the roles of miRNAs in the regulation, correlations, and diagnosis of glioblastoma. It would be a nice review if the following could be improved by the author during revision:
1. None of the tables were referred to / cited in the text;
2. The 1st title Classification of GBM (line 52) should be modified a little to mention miRNAs as well;
3. The abbreviation ‘MGMT” appeared very early in the abstract (line 24) and the introduction section (started from line 119), but the term was only explained in detail on line 519. It should be explained earlier in the article;
4. The authors should consistently use “GBM” as the abbreviation for glioblastoma throughout the text;
5. The 4th and 5th titles (line 181 and line 211) are both related to the 3rd title miRNA (line 167). They could be combined together;
6. The title of Fig. 1 MicroRNA biogenesis and therapeutic applications (miRNAs) (line 202) should be changed since this figure has nothing to do with the therapeutic application. Also, the (a) to (d) marks mentioned in the legend are absent in the figure.
7. The miRNAs mentioned in section IV should be marked consistently as done on pages 16-17;
8. The 8th title miRNA as a biomarker (line 596) and the title of Fig. 5 Dysregulated miRNA biomarkers need to be rewritten to accurately reflect the content, since none of the miRNAs mentioned was a biomarker under clinical use or in clinical trials;
9. Some references are mislabeled in the text;
10. The authors should clarify that miRNA-based replacement therapy (pages. 18-19) and oligonucleotide therapy (page. 19) are still candidate approaches but not yet used for therapy. The titles of Fig. 2 and Fig. 3 (lines 785 and 787) should be changed accordingly.
Reviewer 2 Report
The manuscript titled A comprehensive review of miRNAs and their epigenetic effects in glioblastoma by Hera Hasan et al. elaborate on known miRNAs linked to glioblastoma, their distinctive oncogenic or tumor suppressor roles, on how epigenetic modification of miRNAs, particularly by methylation, leads to their upregulation or downregulation in glioblastoma, and we will try to identify those miRNAs those are the potential regulators of MGMT expression and their role as predictors of tumor response to temozolomide treatment and of survival in temozolomide treated patients. We will also comprehensively discuss known miRNA-based signatures to be used as novel diagnostic and therapeutic markers for glioblastoma.
The authors summarize nearly 100 articles on the subject and produce a very informative and comprehensive manuscript. The version I received for review still contains some editing problems that have made the review more laborious.
I think the content is good and well organized, although a good review of English is mandatory, regarding the figures:
Figure 1 is trivial and at the level of any textbook, it could be either improved or deleted.
Figures 2 and 3 could be joined in a more elegant and less crude horizontal.
My version of Figure 4, Overview (Schematic Diagram), is a mess, there are undefined acronyms (HK, LDHA...) - all need to be defined.
The overview figure should be clear in a length of no more than one page, and this one takes up 3.
The manuscript titled A comprehensive review of miRNAs and their epigenetic effects in glioblastoma by Hera Hasan et al. elaborate on known miRNAs linked to glioblastoma, their distinctive oncogenic or tumor suppressor roles, on how epigenetic modification of miRNAs, particularly by methylation, leads to their upregulation or downregulation in glioblastoma, and we will try to identify those miRNAs those are the potential regulators of MGMT expression and their role as predictors of tumor response to temozolomide treatment and of survival in temozolomide treated patients. We will also comprehensively discuss known miRNA-based signatures to be used as novel diagnostic and therapeutic markers for glioblastoma.
The authors summarize nearly 100 articles on the subject and produce a very informative and comprehensive manuscript. The version I received for review still contains some editing problems that have made the review more laborious.
I think the content is good and well organized, although a good review of English is mandatory, regarding the figures:
Figure 1 is trivial and at the level of any textbook, it could be either improved or deleted.
Figures 2 and 3 could be joined in a more elegant and less crude horizontal.
My version of Figure 4, Overview (Schematic Diagram), is a mess, there are undefined acronyms (HK, LDHA...) - all need to be defined.
The overview figure should be clear in a length of no more than one page, and this one takes up 3.
Reviewer 3 Report
Dear authors,
Thank you for providing the opportunity to review the manuscript. In essence, the overview of miRNAs and their epigenetic effects in GBM is quite well summarized. However, a few concerns need to be addressed before the manuscript can be accepted for publication:
1. The abstract should capture some method/results or main findings that give an overview conclusion. The relation of epigenetic-drug resistance-GBM progression must be clearly described in the abstract and introduction.
2. The classification of GBM is outdated and the histology and grading. Please revise this whole section based on the latest CNS5. Please also ensure the works reviewed in this review manuscript conform to the IDH-mutation status (even at the cell lines level).
3. Which miRNAs play / prospective target in epigenetics and the basis of this with clinical correlation? The relationship of miRNAs with epigenetics changes and also how they lead to TMZ resistance should be discussed clearly, and their expression to be correlated with clinical relevance and survival analysis (based on IDH mutation, epigenetic statuses such as MGMT methylation vs unmethylation, etc).
5. Additionally, do they play different roles and epigenetic mechanisms differently, in different GBM subtypes? And what if the epigenetic regulations other than just looking at TMZ?
4. Unsure why there is 'PTEN or p53??' - line 260
5. The section's challenges and limitations are too oversimplified and to some extent quite superficially discussed. Certainly when discussing epigenetics with regards of miRNAs, various issues from preclinical to clinical in term of epigenetics itself play a major point of discussion. To some extent, the review should also look into the different epigenetics regulation pre- and post-therapy issues, and also to review some work on this, even at preclinical levels.
6. The conclusion does not capture the real essence of the review analysis. There is no requirement for authors to restate the known facts of GBM in GBM. Additionally, most conclusion points are oversimplified without stating the main current findings and analysis.Furthermore, statements such as "To improve the efficacies of various anti-tumour therapies, efforts should be done to identify prognostic markers for intratumour heterogeneity which will aid in predicting 815 and in improvement of therapeutic results. To maximize the efficacy of oligonucleotide therapy, nanoparticles as a miRNA delivery system can be employed and, in future, they constitute a promising therapy in the treatment of GBM' - sound very superficial and do not add scientific knowledge, especially when such facts are known in the open.
Moderate editing of English language
Round 2
Reviewer 1 Report
NA
Author Response
thanks
Reviewer 2 Report
The authors have made most of the suggested changes point by point. However, it still contains a series of topical phrases and general winding-rambling on that makes the reading to take more time than necessary.
Like epeating, such as:
"TMZ is a chemotherapeutic drug which is currently in use as a first choice of treatment of high-grade glioma [57]. It is an alkylating agent and it induces its anti-tumor effect by methylating DNA at the O6 and N7 position of guanine and the N3 position of adenine [60], causing cytotoxicity and killing of cancerous cells. Stupp et al., claims that TMZ therapy in addition to radiation therapy increases overall median survival to 2 months [61 ]. Also, surgical resection and radiotherapy, when combined with TMZ increased median survival from 12.1 to 14.6 months.
Even in the conclusions trivial paragraphs:
"GBM is a lethal brain tumor with very poor prognosis. The median survival time of patients with GBM is 14.6 months and despite discovery of various anti-cancer therapies, complete cure of GBM is still impossible...
Normally, any GBM reader knows that before starting the article. Which makes the article longer and unreadable than usual and seems more like a thesis introduction than a review on "miRNAs and their epigenetic effects"
Figures 1 and 2 (of the new version) are oversized, and not that useful.
Reviewer 3 Report
Dear Authors,
Thank you for the revision.
Although the authors have done the revision to the best of their ability, the major concern is the classification of GBM.
As it is with the latest classification, GBM is no longer considered to have IDH mutation or secondary. Please refer to here - https://doi.org/10.1093/neuonc/noab106
This is a major issue and thus, the pieces of literature used, data used and analyzed in the revised manuscript must adhere to this. Please re-analyze the analysis done, and carefully select the relevant literature/references.
While doing so, please revise the description of GBM classification.
Moderate editing is needed but should be fine during proofreading after acceptance (following the revision).
